# Combatting Visual Fake News with a Professional Fact-Checking Tool in Education in France, Romania, Spain and Sweden

Thomas Nygren [1,*], Mona Guath [1,2], Carl-Anton Werner Axelsson [1,3] and Divina Frau-Meigs [4]

1   Department of Education, Uppsala University, 750 02 Uppsala, Sweden; mona.guath@psyk.uu.se (M.G.);
    carl-anton.werner.axelsson@it.uu.se (C.-A.W.A.)
2   Department of Psychology, Uppsala University, 751 42 Uppsala, Sweden
3   Department of Information Technology, Uppsala University, 751 05 Uppsala, Sweden
4   Digital Humanities, University Sorbonne, Nouvelle, 75006 Paris, France;
    Divina.Frau-Meigs@sorbonne-nouvelle.fr
*   Correspondence: thomas.nygren@edu.uu.se

**Abstract:** Educational and technical resources are regarded as central in combating disinformation and safeguarding democracy in an era of 'fake news'. In this study, we investigated whether a professional fact-checking tool could be utilised in curricular activity to make pupils more skilled in determining the credibility of digital news and to inspire them to use digital tools to further their transliteracy and technocognition. In addition, we explored how pupils' performance and attitudes regarding digital news and tools varied across four countries (France, Romania, Spain, and Sweden). Our findings showed that a two-hour intervention had a statistically significant impact on teenagers' abilities to determine the credibility of fake images and videos. We also found that the intervention inspired pupils to use digital tools in information credibility assessments. Importantly, the intervention did not make pupils more sceptical of credible news. The impact of the intervention was greater in Romania and Spain than among pupils in Sweden and France. The greater impact in these two countries, we argue, is due to cultural context and the fact that pupils in Romania and Spain learned to focus less on 'gut feelings', increased their use of digital tools, and had a more positive attitude toward the use of the fact-checking tool than pupils in Sweden and France.

**Keywords:** fake news; media and information literacy; teaching and learning; fact-checking; lateral reading

## 1. Introduction

Faced with the challenges that are caused by information disorder and infodemics, there is a demand for educational interventions to support citizens and safeguard democracy [1–3]. Education is considered to be key, since automated fact-checking has significant limitations, not least when it comes to debunking visual images and deep fakes [4,5]. In addition, platform companies and fact-checkers struggle to keep pace with the speed and spread of disinformation (e.g., [1,6]), which makes it critical that citizens develop resilience to disinformation by learning to navigate digital news in more up-to-date and autonomous ways.

Disinformation—defined as inaccurate, manipulative, or falsified information that is deliberately designed to mislead people—is intentionally difficult to detect. This poses a challenge not only for professional fact-checkers in mainstream media and digital platforms, but also for media literacy specialists, whose expertise does not extend much beyond imparting basic source verification strategies [3]. Yet, the journalistic profession has been able to benefit from a growing number of fact-checking initiatives that have generated digital tools and novel responses to infodemics. However, such tools have not broadly reached the general public, which has mostly been left to its own devices. This gap between professionals and the general public is further widened by the evolution of

disinformation itself; fake news is now not only text-based, but also increasingly image-based, especially on the social media used by young people, and so debunking news requires more sophisticated approaches.

Building resilience to fake news requires navigating online information in new ways and with the support of digital tools, similar to the methods used by professional fact-checkers [7–9]. Because new technology makes it hard to see the difference between a fake and a real video [10] or to distinguish a misleading image in a tweet from a credible one [11], teenagers often struggle to determine the credibility of images and videos when these are presented in deceptive ways [12–14]. Citizens need a combination of digital knowledge, attitudes, and skills to navigate the complicated digital world of post-truth, as highlighted by theories of media and information literacy, such as transliteracy [15] and technocognition [8].

Young people growing up in an era of online misinformation have been found to struggle to separate fake news from real news [12,14,16–18]. Teenagers stating that they are quite skilled at fact-checking may not hold the skills they think they have [13,19]. The idea that young people are digital natives, knowing how to navigate digital media much better than other generations, does not have any support in the research. Instead, there is a call for educational efforts to promote the media and information literacy of teenagers with diverse backgrounds [14,20,21].

Research has highlighted the existence of a media and information literacy divide between people and pupils in different groups, highlighting a digital inequality between citizens [14,22–25]. Teenagers with poor socio-economic status may spend more time online on entertainment and simple tasks than peers with better support from home, and they may also find it difficult to separate fake news from real news [14,21,26,27]. Access to computers will not automatically bridge this divide since source-critical thinking has multiple interlinked dimensions and it is very complex and intellectually challenging to determine whom to trust online [28,29]. Pupils need more education designed to promote media and information literacy in general and visual transliteracy in particular in order to overcome this divide in different contexts.

Research indicates that it is possible to support people's abilities to evaluate on-line information by giving short instructions on how to identify misleading headlines on Facebook and WhatsApp [7], by the use of games that are designed to alert against manipulative tweets [30] and by educational interventions that support pupils' civic online reasoning [20,21,31]. However, because the technological advances in visual media manip-ulation are leveraging the spread of false or misleading information, researchers are calling for 'more intensive digital literacy training models (such as the "lateral reading" approach used by professional fact checkers)' ([7], p. 7).

In this study, we took on this challenge by evaluating a professional digital fact-checking tool in classroom settings in France, Romania, Spain, and Sweden. The aim of this design-based study was to make the professional plug-in InVID-WeVerify useful in curricular activity in order to improve pupils' skills in evaluating misleading images and videos. We investigated the potential benefits and challenges of implementing the latest advances in image and video verification in collaboration with teachers across Europe.

The tool, InVID-WeVerify, is a free verification plug-in that is available in multiple languages used today by professional journalists and fact-checkers to verify images and videos in newsrooms, such as France24, India Today, Canal 1, and Volkskrant [32]. The plug-in has been downloaded across the globe more than 40,000 times, and it is used on a daily basis by, among others, fact-checkers at Agence France-Presse (AFP) to investigate rumours and suspicious content regarding, for example, Covid-19 and politics.

## 1.1. Educational Interventions to Support Fact-Checking in a Post-Truth Era

International organizations, like UNESCO and the European Union, underscore the im-portance of education to promote so-called media and information literacy as an important defence against propaganda and disinformation [1,33]. Media and information literacy may

be viewed as an umbrella term covering knowledge, skills, and attitudes described by researchers as information, news, media, and digital literacies [33–35]. Information literacy—the ability to evaluate and use information wisely—has especially been noted as a 'survival skill' [36]. In line with this, the theory of civic online reasoning underscores how "the ability to effectively search for, evaluate, and verify social and political information online" is essential for all citizens ([18], p. 1). The multi-modal aspects of digital information involve new challenges when people search for, find, review, analyse, and create information [37–39]. Researchers also call for more research on civic online reasoning with new and more complex tasks and test-items paying attention to pupils' knowledge, skills, and attitudes in different educational settings [20].

Today, the ability to read, write, and interact across a range of platforms, tools, and media, described as transliteracy, has become a key literacy in a world of digital multimodal information [40]. Transliteracy has been enlarged to embrace the double-meaning of digital convergence: '1. the ability to embrace the full layout of multimedia, which encompasses skills for reading, writing, and calculating with all the available tools (from paper to image, from book to wiki); 2. the capacity to navigate through multiple domains, which entails the ability to search, evaluate, test, validate and modify information according to its relevant contexts of use (as code, news, and document)' ([41], pp. 15–16). Transliteracy echoes technocognition as an emerging interdisciplinary field that involves technological solutions incorporating psychological principles to solve disinformation issues [8]. In the present study, we focus on the latter aspect of transliteracy, more precisely, how tools can facilitate navigating a digital information landscape.

Scholars point out that journalistic principles and technology may support citizens in navigating a digital world of deep fakes and misleading visual and text-based information [8,9]. The use of digital tools to support the verification of news has been discussed in terms of technocognition and civic online reasoning [42]. These prescriptive theories emphasise that citizens need to be better at scrutinising online information, and that this is a psychological, technological, and educational challenge. In a post-truth era, people need to consider that images and videos may be manipulated and also be able to use digital resources, such as text search and reverse image search, to corroborate information. Professional fact-checkers use technology to read laterally, which means that they verify information on a webpage by corroborating it with information on other credible webpages [9]. Researchers note that education and technology that support lateral reading may be key to safeguarding democracy in a post-truth era that is saturated by disinformation [7–9]. However, the use of professional fact-checking tools in education to support pupils' transliteracy, lateral reading, and technocognition has not been studied in previous research to date.

What has been noted in media and information literacy research is that pupils often struggle to separate credible from misleading digital multimodal information [12,14]. Even individuals with proficient news media knowledge may struggle to evaluate evidence online [17,43]. The high expectations of news literacy programmes [3] should be understood in light of these challenges. Scholars also emphasise that technology and educational interventions are not quick fixes for the complex challenge of misinformation [44]. More time in front of computers does not necessarily make pupils more skilled at navigating online information [18,45,46]. Without adequate media and information literacy, pupils may fail to separate credible information from misleading informationl, because they are not able to use effective and adaptive strategies when evaluating manipulated images and junk news [28]. In education, it is critical that the educational design includes a combination of challenging and stimulating tasks, and different types of hard and soft scaffolds to help pupils use online resources in constructive ways [47–52].

While noting the many challenges, we still find a few studies highlighting the ways in which it is possible to support pupils lateral reading skills in education. Educational designs for promoting civic online reasoning have made it possible for teenagers at the university and high school level to scrutinise digital news in a similar manner to professional fact-checkers [20,21,31,53]. Previous research has also identified that it is possible for upper

secondary school pupils to use digital tools that are designed for professional historians in critical and constructive ways if they are supported by an educational intervention comprising supporting materials and teaching [54,55].

### 1.2. Design-Based Research

Implementing innovative technology in schools is often linked to design-based research in education, also known as design experiments, design research, or design study [56]. Testing and developing digital tools that may hold new dimensions and practices is often at the core of design-based research [56], not least since this may provide new practical and theoretical insights. design-based research aims to "test and build theories of teaching and learning, and produce instructional tools that survive the challenges of everyday practice" ([57], p. 25). The usefulness of design-based research comes from the methods where researchers and teachers collaborate to identify challenges and test new materials and methods in complex classroom settings with a purpose to promote pupils' learning [58,59]. In line with a previous call for "research that features practitioner co-creation of knowledge as a vehicle for use and uptake" ([60], p. 98) and congruent with Akkerman et al. [61], we acknowledge that dialogue with teachers regarding technology is essential in educational design-based research.

Design-based research advocate Ann Brown [62] stresses the importance of collecting data from messy classrooms in order to measure learning where it usually occurs. She underscores how measuring effects through pre- and post-tests designed to fit the research focus and this is central in our study ([62], Additionally, see Figure 1 and the section below on materials and methods). Our research is based on the assumption that the design of materials and methods is important for learning and we focus on developing new tools and theories for teaching in the complex reality of teaching and learning [63]. In line with the methodology of design-based research we see that the materials and methods that were developed through iterative studies in the classrooms should preferably survive the challenges of classroom practices and remain to be used in teaching long after the research project is completed [57,64]. Thus, design-based research professionals argue that educational science needs to develop ideas and products that work in thoughtful ways [65] and, in this article, we present some steps in this direction.

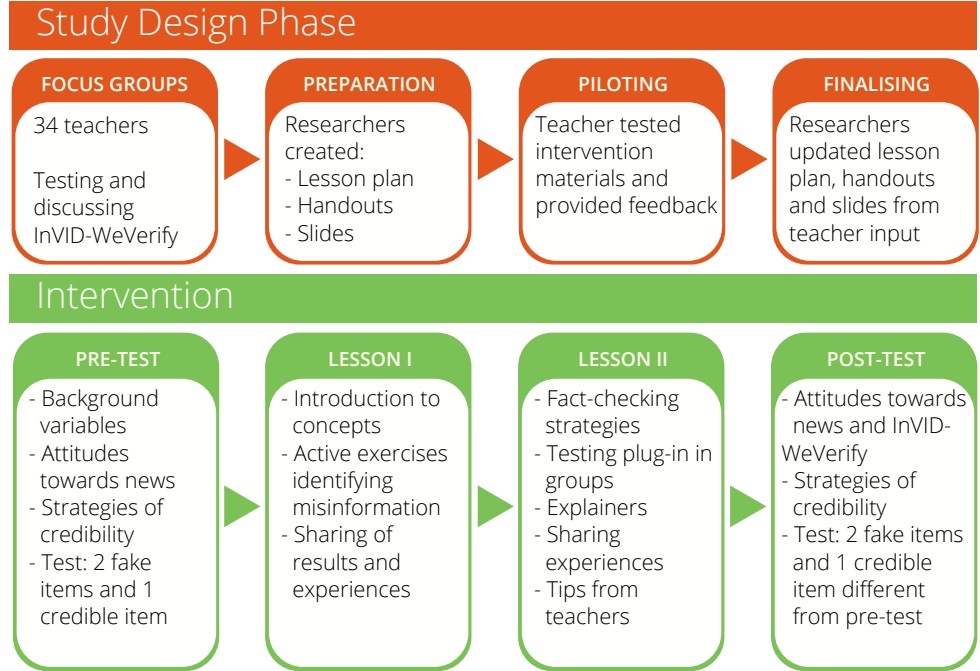

**Figure 1.** Study design.

*1.3. The Present Study*

Noting the major challenge of fake news, the limited ability of pupils to navigate information in digital environments and the existence of digital fact-checking tools, we see an opportunity to answer the calls for media and information literacy interventions with a design-based research approach. In this study, we investigated whether a two-hour educational intervention in four European countries, using a computer-based tool that was designed for professional fact-checkers, could stimulate pupils' visual literacy and make them better at determining the credibility of digital news. We explored the following hypotheses:

**Hypothesis 1.** *Pupils will become more skilled at assessing digital news after the intervention. Specifically, we expected the following:*

*a*    *The total post-test score will be significantly better than the total pre-test score.*
*b*    *The ability to debunk false and true items separately will also be significantly better in the post-test.*

**Hypothesis 2.** *Better performance in assessing information will be facilitated by the use of digital tools.*

In addition, we investigated the following exploratory research questions:

**Q1**    How do media attitudes and digital information attitudes vary across countries?
**Q2**    How does performance on pre- and post-test vary across countries?

## 2. Materials and Methods

*2.1. Participants*

A total of 373 upper secondary school pupils, aged 16–18, across the four countries participated in the lessons and responded to the questionnaire during the Autumn term of 2020. All of the pupils agreed to complete the pre-test with anonymised responses for research purposes (with informed consent in line with the ethical guidelines of all countries). Of 373 pupils, there were 238 who took both the pre- and post-test, and this was the number of participants that we used in the analyses. The number of complete responses in each country was: 59 in France, 22 in Romania, 47 in Spain, and 110 in Sweden. The gender distribution was: 144 girls, 83 boys, and 11 pupils, which indicated that they did not wish to specify their gender or identified as non-binary. The different sample sizes in each country were primarily due to lockdowns and challenges that are linked to schooling during the Covid-19 pandemic.

*2.2. Material*

Media and information literacy theories, such as transliteracy, civic online reasoning, and technocognition, all emphasise the importance of using digital tools when evaluating online information. The plug-in InVID-WeVerify is such a tool and it offers multiple functionalities that provide support to users when verifying videos and images [66,67]. InVID-WeVerify makes it possible to (a) conduct reverse image searches with multiple search engines (e.g., Google, Yandex, Bing); (b) analyse images with forensic filters to detect alterations in their structure, such as quantisation, frequencies, colours, and pixel coherence; (c) scrutinise the details of an image with a magnifying lens; (d) fragment videos into key-frames; and, (e) retrieve metadata about videos and images.

Thus, the tool is designed to support advanced media verification strategies. However, introducing a professional tool for fact-checking in education may have little effect if the tool is not understood or is found to be unusable by teachers or pupils. General media literacy principles reinforce the need for sense-making uses of technology in terms of knowledge acquisition and societal values, recommending that the tool or operational device not be the main entryway to problem-solving [41]. This is consistent

with prior research pointing to the fact that dialogue with teachers regarding technology is essential in educational design-based research [61]. Therefore, we initiated our endeavour with a study design phase, in which 34 teachers from France, Romania, Spain, and Sweden participated in focus group discussions with the aim of testing the tool and providing feedback on the usefulness of the tool in education (for a complete design overview, see Figure 1). Focus group discussions were organised to assess the perception of disinformation by teachers in their local context and their perspectives on InVID-WeVerify functionalities, especially in relation to image reverse search, automated video key frames extraction, and image forensics. The findings from these focus group discussions highlighted that implementing the tool in class may be very complicated and pointed to a need for scaffolds [68]. The results also highlighted some cultural differences; for instance, challenges may be greater in Romania than in Sweden due to the different media cultures and technical resources available in the two countries. Learning from teacher feedback, we designed educational materials to scaffold the use of the tool in classrooms. This was achieved through close collaboration between teachers and researchers. Researchers from the participating countries discussed and created materials and methods for stimulating transliteracy and technocognition. These materials were then introduced to a teacher who tested and piloted them in teaching and then provided feedback. In this phase, we also designed and piloted credible and fake news items for use in pre- and post-tests (see example, items in Appendix A). The final educational design, limited to a two hour intervention, was agreed upon by 16 social studies teachers and eight researchers that were situated in the four countries.

Materials for the educational intervention included a lesson plan for teachers, handouts for use in the classroom, and presentation slides. The educational intervention was introduced with an initial 60 min lesson on the theme 'News in a world of fake news: Definitions, credibility, and identification of different types of misinformation', and included a combination of lectures (presenting concepts that are linked to misinformation, examples of fake news, and summing up discussions) and active exercises for pupils where they were asked to (a) come up with news sources (see Figure 2) and (b) identify different types of misinformation. The lesson was concluded with a sharing of results and collective discussions about what the pupils learned from the lesson.

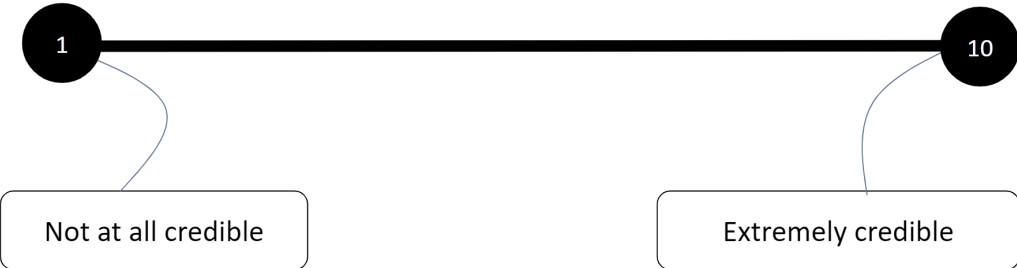

**Figure 2.** Pupil active group task designed to stimulate conversation and link the educational content to pupils' perceptions of news.

The second lesson, which was also 60 min in duration, focused on 'Individual defence against misinformation and disinformation: Understanding image manipulation with InVID-WeVerify'. This lesson started with a short lecture on how fact-checkers use lateral reading and verify information by considering: (a) Who is behind this information? (b) What is the evidence? and (c) What do other sources say? [9]. The teacher acted as a fact-checker by conducting a reverse image search and forensic analysis of an image with InVID-WeVerify. Next, the teacher showed a fake video and verified this using the InVID-WeVerify key frames analysis. Thereafter, the pupils downloaded the plug-in and worked

in groups of two to three to verify images and videos with InVID-WeVerify. The pupil task (Figure 3) was provided in PDF format, which makes it possible for them to click on hyperlinks and use InVID-WeVerify 'in the wild' with authentic examples of misleading images and videos.

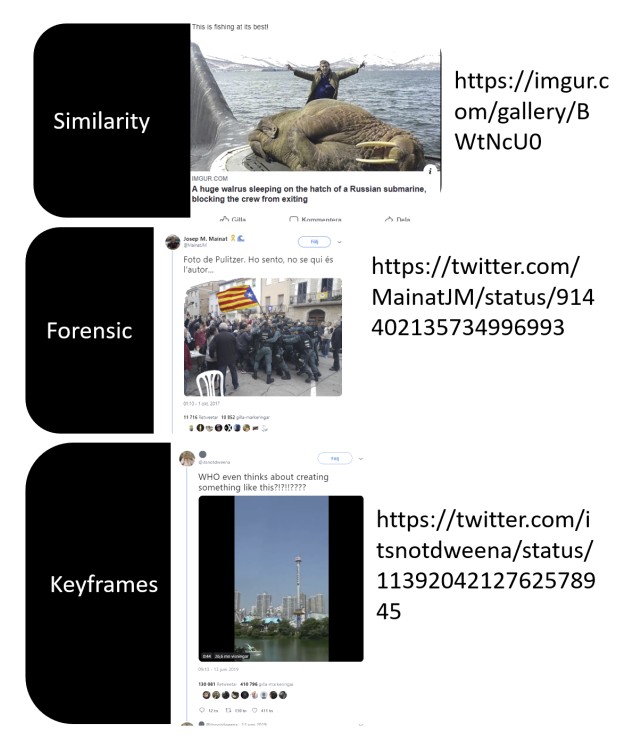

**IS IT CREDIBLE, BIASED OR FAKE?**
Who is behind this information?
What is the evidence?
What do other sources say?

**Pupil task (2-3 people in each group)**
1) Similarity: Double check! Reverse image search to see where the image comes from and possible manipulations. Explore several search engines, like Yandex, Bing and Google.

- Start by clicking this Imgur post

Right-click the image and use Fake video news debunker by Invid – to do reverse image search in Yandex etc. Is it credible? Why/why not?

2) Forensic: You are the detective! Spot what may have been manipulated.

- Start by clicking this Twitter post

Right-click the image and use Fake video news debunker by Invid – Forensic. Submit & scroll down. Is it credible? Why/why not?

3) Keyframes: Stop and analyze the video!

- Start by clicking this Twitter video

Use keyframe tool in Invid. Insert twitter link in keyframe. Use magnifier to look at name of carosel. Then do a reverse image search. Is it credible? Why/why not?

https://imgur.com/gallery/BWtNcU0

https://twitter.com/MainatJM/status/914402135734996993

https://twitter.com/itsnotdweena/status/1139204212762578945

**Figure 3.** Pupils' task designed to stimulate and scaffold their use of InVID-WeVerify.

The step-by-step task was designed to prompt pupils to use different techniques to debunk images and videos. After the group work, the teacher showed slides explaining how to debunk the fake images and video using InVID-WeVerify (see Figure 4) and discussed, in class, what the pupils had learned. Summing up the two lessons, the teacher then presented some information regarding how to navigate online news in clever ways with tips, such as: (a) set filters on credible sources (i.e., reviewed news feeds from established news media); (b) be careful about frames and deceptive design (what looks great may be the very opposite); (c) rely on several search engines; (d) look for other independent sources, double check!; (e) think first then share—share with care!; and, (f) stay cool!

### 2.3. Procedure

With an aim to develop new methods and materials that are useful in the complexity of everyday classroom practices, we made sure to collect a rich set of data, enabling us to investigate the possibilities and challenges of this educational design [56,64]. The intervention took place in October 2020 during the Covid-19 pandemic, presenting us with a special challenge in conducting the educational effort. It was initially planned for March 2020, but was postponed due to school lockdowns in three out of the four countries. The intervention started and ended with an online questionnaire, which included a test with two fake news test items (a misleading image and a misleading video) and one credible news test item. We made sure to include both credible and fake information, because scholars have noted that exposure to fake news may lead to diminished levels of trust in credible news [69]. We used different items in the pre- and post-tests, and counterbalanced these items between groups to ensure that the results would come from the intervention and not the test items. Test items—one true news item, one item with a manipulated image,

and one fake video—were introduced with the following instruction: 'You will now be asked to rate the credibility of news on social media. When you do this, you are free to double check the information online'. The items were presented in a randomised order to minimise the order effects. All of the items included were social media posts intended to 'go viral', which is, they were designed to attract attention, clicks, and shares.

**EXPLAINERS**

1) Walrus was on a Russian submarine undergoing repairs in 2006.

The popular image of young man is fake. (Snopes, 2019)

*Click bait!*

2) Image is from protests.

Flag is inserted to look more iconic (El Pais, 2017)

*Ideological misinformation!*

3) Video is fake. Producer is unknown. (Metro, 2019; Wired 2019)

*Click bait!*

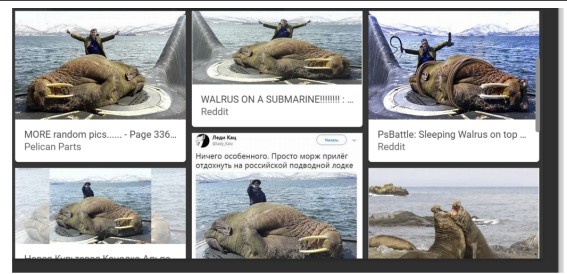

**Tips & tricks!**
Bing reverse image search + Google key word search "walrus submarine fake photo" scroll down to Snoopes article

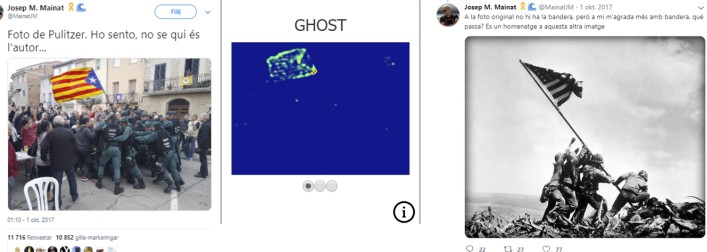
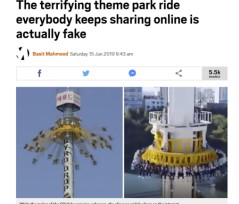

**Tips & tricks!**
Forensic + Google reverse image search "Catalonia fake" – scroll down to El Pais article in hitlist

**Tips & tricks!**
Keyframes + Google reverse image search + Magnifier to see "Gyro drop". Text search "Gyro drop fake" scroll down to Wired article in hitlist

**Figure 4.** Explainers informing how to debunk fake images and video using InVID-WeVerify.

The pre- and post-test also included questions regarding the pupils' use of digital tools when they assessed the credibility of the information. We asked: 'When you answered the questions in the survey, did you use any digital tools? For example, did you use Google or reverse image search to examine the credibility of the news? Yes/No'. If they checked 'Yes', we asked them: 'How did you use digital tools to help you examine the credibility of the news? (check multiple boxes if you did multiple things) (a) I used text searches (for instance on Google), (b) I did reverse image searches, (c) I used multiple search engines, and/or (d) other (please specify)'.

We also asked the pupils questions regarding their background, their attitudes towards news, and how they usually determine credibility (see Appendices B and C). These factors have been identified as important in previous research and in research highlighting the complexity of media and information literacy [13,14,70]. In order to investigate the pupils' self-perceived skills and attitudes, we asked them to rate their ability to find and evaluate information online and their attitude towards credible information sources in line with previous research [13,14]. Answers were given on a five-point scale (see Appendix B Question 3–6). The participants were then asked to rate statements on their strategies to determine the credibility of news on a scale from 1 (never) to 7 (very often), with questions being adapted from Frunzaru and Corbu [70] (see Appendix B Question 7).

In the post-test, we asked the pupils to rate their user experience of InVID-WeVerify in order to assess their perception of the visual verification tool. All of the questions in the tests were asked in the native language of the pupils. We also interviewed teachers and asked them (a) what worked well in the teaching, (b) problems in the teaching, and (c) suggestions for improvements.

*2.4. Design*

The study design was a repeated measurement using pre- and post-tests around the educational intervention with the InVID-WeVerify-tool, where the order of the pre- and post-test items were counterbalanced to avoid the order effects.

*2.5. Analysis*

We transformed false item scores by subtracting each false score from the maximum rating for each false item, so that a high score signified good performance. We also reversed the items in the news evaluation test, where higher ratings indicated less awareness of the need to fact-check information. We then summed all of the pre- and post-test items for the respective order conditions to yield two scores, a pre-test score and a post-test score, respectively.

We made a two-way mixed ANOVA with time as the repeated measure, and the use of digital tools on post-test and language as the between-subjects variable, and total test score as the outcome variable, in order to analyse performance in relation to the hypothesised relationships. Because the sample sizes in each country were unequal, we followed Langsrud's advice [71] and made ANOVAs with Type II squares for unbalanced design. Essentially, the Type II sum-of-squares allows for lower-order terms to explain as much variation as possible, adjusting for one another. Because of non-normally distributed data, we analysed the post-test scores on false and true items with the Wilcoxon rank-sum test, with the use of digital tools as the independent variable.

Next, we investigated how the attitudes differed between the countries. Summary statistics of all attitudes are provided in Tables A1–A11 in Appendix C; only statistically significant differences are discussed here in the text. For the self-rated skills and attitudes in relation to digital proficiency, we ran one-way ANOVAs using the Type II sum-of-squares, with language as the independent variable. For the self-rated attitudes towards news and attitudes towards the digital tool, we made pairwise t-tests for each attitude, separately rating pre- and post-tests for each language.

**3. Results**

*3.1. Performance Test*

The maximum total score on the performance test was 21, with each item providing a maximum score of 7. The mean total score on pre-test across all countries was 12.2 ($SD = 3.2$), and the mean total score on post-test was 13.2 ($SD = 2.9$), a statistically significant difference ($t(467.63) = 3.58, p < 0.001$). Table 1 presents the pre- and post-test performance for each language version of the tests. The median total score on false items on pre-test was 9 ($MAD = 3.0$) and 10 ($MAD = 3.0$) on post-test, a difference that was statistically significant ($W = 34742, p < 0.001$). For the true items, the median on the pre-test (3; $MAD = 3.0$) did not result in a statistically significant difference ($W = 26,283$, $p = 0.29$) from the median on the post-test (3; $MAD = 3.0$).

**Table 1.** Mean total scores on pre- and post test for the separate languages with standard deviation in parentheses.

| Measure | French | Romanian | Spanish | Swedish |
|---|---|---|---|---|
| Pre-test score | 12.1 (3.2) | 10.5 (4.5) | 12.1 (3.1) | 12.5 (2.8) |
| Post-test score | 13.1 (2.6) | 13.7 (3.4) | 13.9 (3.4) | 12.7 (2.7) |

*3.2. Differences in Pre- and Post-Test Scores in Relation to Use of Digital Tools and Language*

Regarding the use of digital tools, 14% of the participants stated that they had used digital tools in the pre-test and 44% stated that they had used digital tools in the post-test when evaluating the news. Table 2 presents the digital tool use in pre- and post-test for each language version of the tests.

**Table 2.** Percentage of digital tool use on pre- and post test for the separate languages.

| Measure | French | Romanian | Spanish | Swedish |
|---|---|---|---|---|
| Pre-test digital tool use | 29% | 23% | 8.5% | 8.1% |
| Post-test digital tool use | 42% | 91% | 64% | 27% |

We ran a mixed $2 \times 2 \times 4$ ANOVA, with time as the within-subjects variable (pre/post), and use of digital tool (yes/no) and language (French, Romanian, Spanish, and Swedish) as the between-subjects variables (for complete results, refer to Table A13 in the Appendix C). The results showed a statistically significant main effect of using digital tools ($F(1229) = 7.29$, $MSE = 11.94, \eta_p^2 = 0.031, p = 0.007$). A post-hoc Bonferonni-corrected *t*-test showed a statistically significantly higher mean total score ($p < 0.001$) with digital tools ($M = 13.1$, 95% CI[12.7, 13.6]) than without ($M = 12.2$, 95% CI[11.7, 12.6]). There was also a statistically significant main effect of time ($F(1, 229) = 18.87, MSE = 6.14, \eta_p^2 = 0.076, p < 0.001$).

A follow-up Bonferonni-corrected *t*-test ($p = 0.008$) showed a statistically significant higher total score on post-test ($M = 13.4$, 95% CI[13.0, 13.8]) than on pre-test ($M = 11.9$, 95% CI[11.5, 12.3]). However, the main effect of time was mediated by an interaction between language and time ($F(3229) = 2.95, MSE = 6.14, \eta_p^2 = 0.037, p = 0.033$; depicted in Figure 5, Panel B).

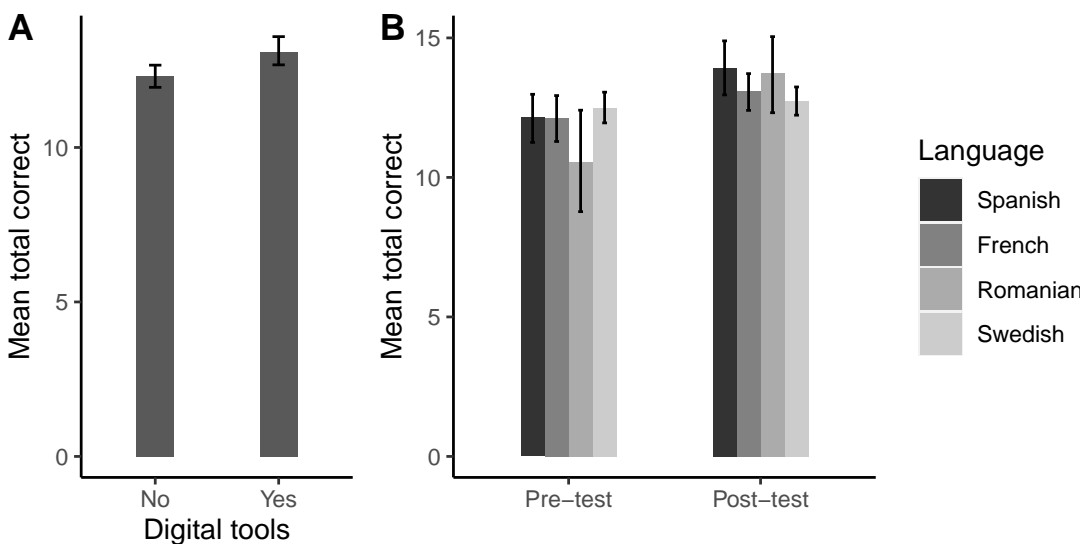

**Figure 5.** Panel (**A**) depicts the mean total score for participants not using/using digital tools in the post-test. Panel (**B**) depicts the mean total score on pre- and post-test for each language. Error bars denote a bootstrapped standard error of the mean.

A follow-up analysis with Bonferroni-corrected pairwise comparisons resulted in a statistically significant difference ($p = 0.021$) between pre- ($M = 12.2$, 95% CI[11.3, 13.0]) and post-test ($M = 13.9$, 95% CI[13.0, 14.7]) for the Spanish language. There was also a statistically significant difference ($p = 0.004$) between pre- ($M = 10.4$, 95% CI[9.24, 11.6]) and post-test ($M = 13.3$, 95% CI[12.1, 14.5]) for the Romanian language. However, the differences between the pre- and post-test for the Swedish and French languages were not statistically significant. To summarise, using digital tools resulted in higher total post-test scores and the post-test scores were statistically significantly higher than the pre-test scores for the Spanish and Romanian languages.

### 3.3. Post-Test Scores on True and False Items When Using Digital Tools

For total scores on false items on post-test, there was a statistically significant difference ($W = 5642, p = 0.028$) with an advantage for those using digital tools. For total scores

on true post-test items, there was also a statistically significant difference ($W = 5383$, $p = 0.007$), with an advantage for those using digital tools. We present medians and median absolute deviations in Table 3. Using digital tools is clearly advantageous; however, there is also greater spread (MAD) in post-test scores within the group using digital tools as compared with the group not using digital tools.

**Table 3.** Medians and median absolute deviation (in parentheses) for scores on total false post-test (maximum 14) and true post-test (maximum 7) for participants using/not using digital tools on post-test.

| Type of Items | No Digital Tools | Digital Tools |
|---|---|---|
| Total false post-test | 10(3.0) | 11(1.5) |
| True post-test | 3(1.5) | 4(3.0) |

*3.4. Attitudes*

For the digital attitude scale, we report the means in Table 4 and, for the news evaluation scale and InVID-WeVerify attitudes, we refer to Tables A1–A11 and Table A12, respectively, in the Appendix C. For the self-rated attitudes and skills, the participants were provided with a five-point rating scale.

**Table 4.** Means and standard deviations (in parentheses) for fact-checking ability, search ability, internet info reliability, and credibility importance based on a five-point rating scale.

| Attitude Measure | $M(SD)$ |
|---|---|
| Fact-checking ability | 3.3(0.9) |
| Search ability | 3.6(0.9) |
| Internet info reliability | 2.8(0.6) |
| Credibility importance | 4.3(0.9) |

The mean scores were moderate for the media and information literacy attitudes, except for internet info reliability and credibility importance. The pupils were quite sceptical about news on the internet (info reliability) and valued credible news highly (credibility importance), which may be due to the fact that they were just about to go through a fact-checking intervention, but it may also reflect a more general scepticism of online information and a positive attitude towards credible news. In addition, pupils rate their ability to assess online information (fact-checking ability and search ability) quite highly.

*3.5. Self-Rated Attitudes and Skills*

For self-rated fact-checking ability (see Figure 6A), there was a statistically significant difference between languages ($F(3234) = 12.36, MSE = 0.72, \eta_p^2 = 0.14, p < 0.001$). Tukey corrected pairwise comparisons showed a statistically significant difference between Spanish and Swedish ($p < 0.001$), as well as between French and Swedish ($p < 0.001$).

For self-rated search ability (see Figure 6B), there was a statistically significant difference between languages ($F(3234) = 17.32, MSE = 0.67, \eta_p^2 = 0.18, p < 0.001$). Tukey corrected pairwise comparisons showed a statistically significant difference between Spanish and Romanian ($p < 0.001$), French and Romanian ($p < 0.001$), Swedish and French ($p < 0.001$), and Romanian and Swedish ($p < 0.001$).

For the ratings of internet info reliability (see Figure 6C), there was a statistically significant difference between languages ($F(3234) = 12.36, MSE = 0.32, \eta_p^2 = 0.039, p = 0.024$). Tukey corrected pairwise comparisons showed a statistically significant difference between French and Swedish ($p = 0.034$). Finally, there were no statistically significant differences ($F(3234) = 1.08, MSE = 0.86, \eta_p^2 = 0.014, p = 0.36$) for the ratings of credibility importance.

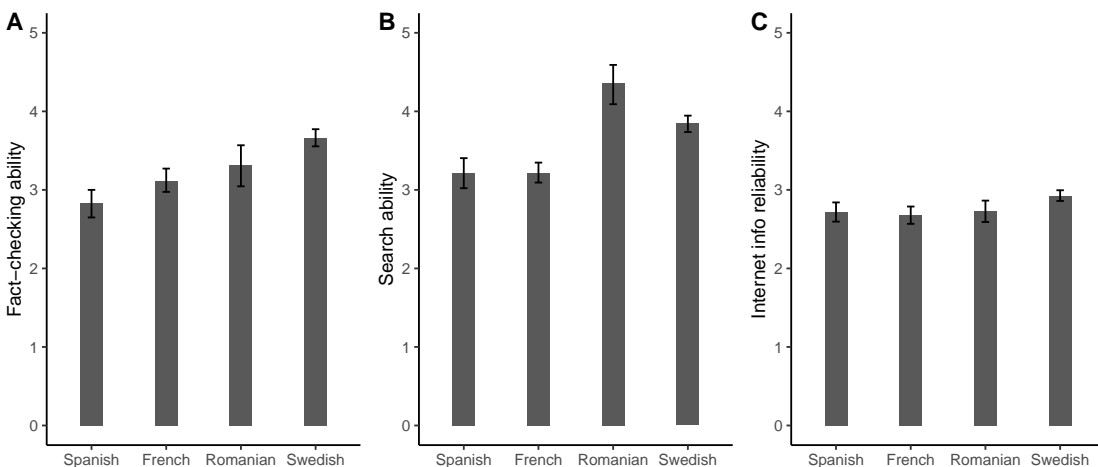

**Figure 6.** Mean ratings (*y*-axis) with standard error of mean for (**A**) fact-checking ability, (**B**) search ability, (**C**) internet info reliability for each language on the *x*-axis.

### 3.6. News Evaluation Ratings

For the news evaluation attitudes, we made pairwise t-tests for pre- and post-ratings separately for each language. There were no statistically significant differences for French.

For Romanian (see Figure 7A), there was a statistically significant difference ($t(40.91) = 3.83$, $p < 0.001$) between pre- and post-test ratings of relying on my gut feeling, with a lower rating on post-test. There was also a statistically significant difference ($t(40.38) = -2.24$, $p = 0.030$) between the pre- and post-test ratings of design of images, with higher ratings at post-test.

For Spanish (see Figure 7B), there was a statistically significant difference ($t(89.48) = 2.92$, $p = 0.004$) between pre- and post-test ratings of 'relying on my gut feeling', with lower ratings at post-test. There was a statistically significant difference ($t(90.81) = -2.49$, $p = 0.015$) between the pre- and post-test ratings of 'search for the source', with higher ratings at post-test. There was a statistically significant difference ($t(89.97) = -2.69$, $p = 0.009$) between pre- and post-test ratings of 'design of images', with higher ratings on post-test.

For Swedish (see Figure 7C), there was a statistically significant difference ($t(215.98) = 2.50$, $p = 0.013$) between the pre- and post-test ratings of 'relying on journalists' reputation', with lower ratings on post-test. There was also a statistically significant difference ($t(209.51) = -4.53$, $p = 0.015$) between pre- and post-test ratings of 'design of images', with higher ratings on post-test.

### 3.7. InVID-WeVerify Ratings

For the ratings on InVID-WeVerify, we present the mean total ratings, with error bars for each language, in Figure 8. There were 17 items in total, each with a 1–7 point rating scale, resulting in a maximum total rating of 119. A higher rating represents a more positive attitude towards the digital tool.

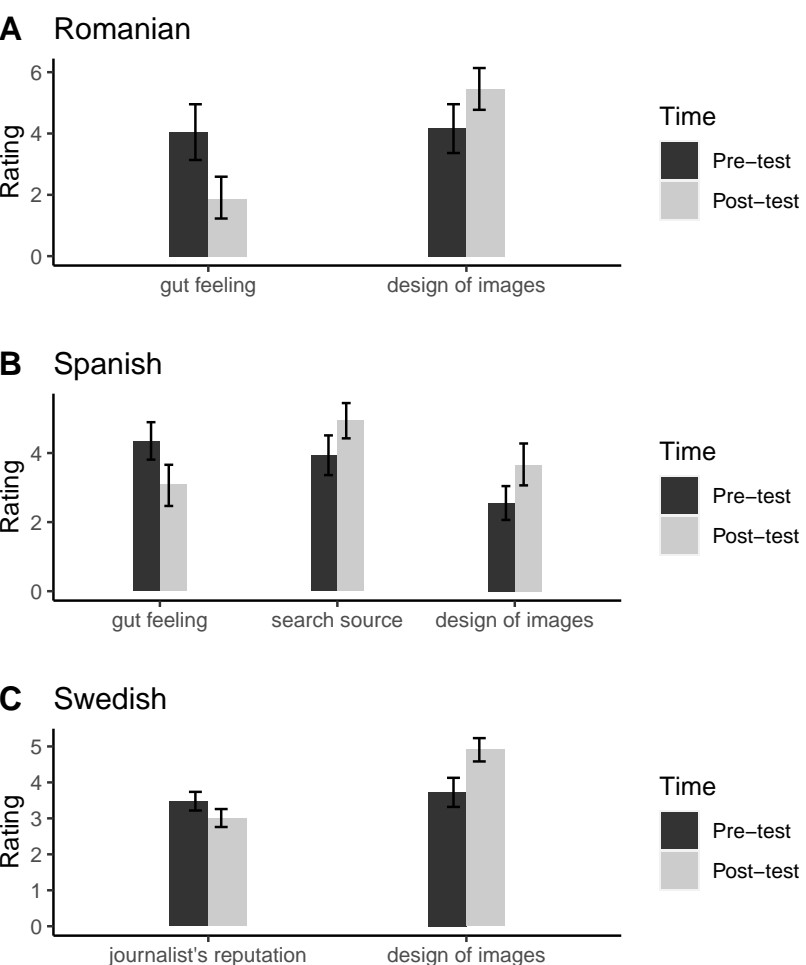

**Figure 7.** Mean ratings (*y*-axis) with the standard error of mean for statistically significant differences between pre- and post-test for items on news evaluation test (*x*-axis) for Romanian (**A**), Spanish (**B**) and Swedish (**C**) language versions of the intervention.

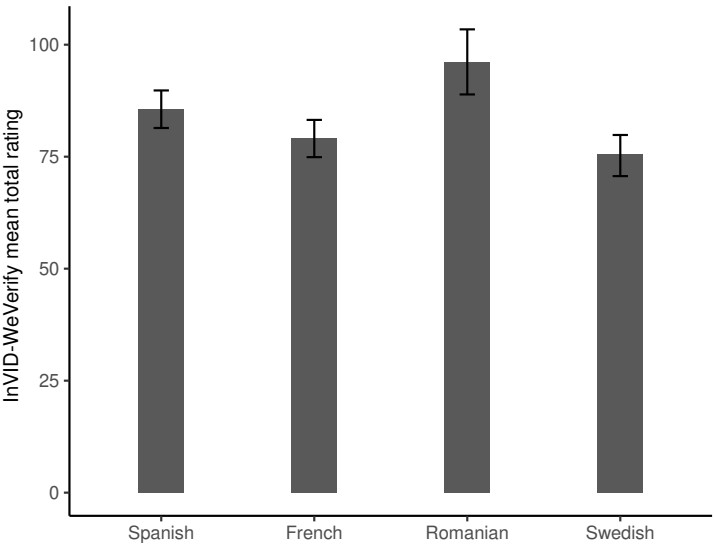

**Figure 8.** Mean total ratings on InVID-WeVerify attitudes (*y*-axis) with standard error of mean (error bars) for each language (*x*-axis).

A one-way ANOVA (Type II sum-of-squares), with language as the independent variable and total mean ratings as the dependent variable, showed a statistically significant main effect ($F(3202) = 8.22$, $MSE = 322.65$, $\eta_p^2 = 0.11$, $p < 0.001$). A follow-up test with Tukey pairwise comparisons showed a statistically significant difference between Spanish and Swedish ($t(202) = 2.98$, $p = 0.017$), French and Romanian ($t(202) = -3.56$, $p = 0.003$), and Romanian and Swedish ($t(202) = 4.52$, $p < 0.001$).

*3.8. Teachers' Impressions from Teaching*

Teachers from all countries found the intervention interesting, but most of them also found it difficult to fit into the limited two-hour time frame. Across countries, teachers emphasised that pupils showed interest in the topic of 'fake news'. Teachers in Spain and Romania, in particular, reported that pupils were very excited about using the new technology in the classroom. In contrast, one of the Swedish teachers reported less enthusiasm than normal in his class. In France, teachers experienced technical difficulties due to problems with installing and using the tool, slow connectivity, and dependency on network quality. Spanish teachers experienced few problems in the implementation and called for an app version of the tool to make it more useful for pupils. Teachers from all countries highlighted the usefulness of the forensic analysis functionality of the tool, and they also reported problems with the analysis of videos functionality (key frame analysis).

*3.9. Summary of Results*

With regard to performance, there were statistically significantly higher mean total scores on post-test as compared with pre-test. Using digital tools on the post-test resulted in higher total scores. Further, for Spanish and Romanian pupils, there were statistically significantly higher scores on total post-test scores when compared with pre-test, but not for French and Swedish pupils. For separate true and false items, there were statistically significant differences between pre- and post-test for false items, but not for true items. Using digital tools produced higher scores on post-test for both false and true items. For the attitude ratings, there were statistically significant differences between languages on media and information literacy attitudes ratings. For fact-checking ability, Swedish ratings were statistically significantly higher than Spanish and French ratings. For search ability, Romanian ratings were statistically significantly higher than all other languages, and there were statistically significant differences between the Swedish and French ratings. For internet info reliability, Swedish ratings were statistically significantly higher than French ratings. For news evaluation ratings, there were statistically significant differences between pre- and post-test ratings for Romanian and Spanish pupils on 'gut-feeling', with lower ratings on post-test. There were also statistically significant differences between pre- and post-test ratings for Romanian, Spanish, and Swedish pupils on 'design of images', with higher ratings on post-test. Finally, there were statistically significant differences between pre- and post-test ratings on 'search source' for Spanish pupils, with higher ratings on post-test; and, statistically significant differences between pre- and post-test ratings on 'journalists' reputation' for Swedish pupils, with lower ratings on post-test.

For mean total InVID-WeVerify attitude ratings, there were statistically significant differences between languages. Romanian speakers displayed higher ratings than French and Swedish, and Spanish speakers displayed higher ratings than Swedish speakers. The attitudes towards the tool are congruent with the reports from teachers, highlighting that pupils in Romania and Spain were especially positive towards the educational intervention with what they perceived as very new and interesting technology.

## 4. Discussion

Previous research has highlighted that digital fact-checking strategies are scarce among historians and students, even at elite universities [9]. However, previous research has also highlighted how it is possible, but also an educational challenge, to support teenagers' abilities to evaluate news [20,21]. In the present study, with regard to our initial hypotheses,

we found that the overall performance on post-test was better than on pre-test (H1a), and that the overall performance on false items was better on post-test than on pre-test, but not on true items (H1b). In line with previous educational interventions that were designed to support teenagers' fact-checking abilities, we saw that not all of the pupils learned to evaluate digital news better. However, we found that our intervention supported pupils in groups with poor performance on the pre-test. This indicates that our intervention with a professional fact-checking tool is possible to implement in education cross-contextually and especially in classrooms with pupils lacking skills to navigate online news. We find that it is possible to stimulate pupils' technocognition and their transliteracy across texts, images, and videos in updated ways. What scholars call for in theory, we test in practice [8,72]. What seem to be complicated cognitive acts in human–computer interaction, using professional fact-checking tools for video and image verification, can be supported in education. Introducing a state-of-the-art fact-checking tool was possible and with the support of the educational design, and not least teachers, it was possible for pupils to better evaluate misleading images and news with the support of technology and training.

In line with our second hypothesis (H2), the results indicate that using digital tools led to better performance on post-test. In the post-test, the pupils performed better on false and true items when using digital tools. Research has previously highlighted how implementing digital tools that are designed for professionals may be very confusing to pupils and hard for them to use without proper support [55]. We do not see this problem in our results, perhaps due to helpful materials and teaching developed in dialogue between researchers and teachers in this design-based research study. Instead, our results indicate that a complicated tool, like Invid-WeVerify, may help pupils to navigate better in the complex world of misinformation. In future research, it would be interesting to investigate more in detail if new technology may actually be more useful to pupils with poor previous research and how technocognition and transliteracy plays out on an individual level in relation to different tools and digital environments. It is evident that using google reverse image search can be useful—but using different search engines can produce different results and being aware of other search engines, such as Bing and Yandex, broadens lateral reading strategies. Conducting image analysis with digital tools may also help pupils to see how algorithms work and how automated fact-checking with forensic analysis can identify manipulated images. In future research, it would be interesting to investigate how pupils' understanding of algorithms and machine learning can be stimulated by using verification tools. The many challenges of misinformation, including misleading images and videos, makes it important to safeguard how citizens hold a rich set of digital tools to support them when navigating online news feeds.

### 4.1. Total Performance Highlights the Importance of Technocognition and Transliteracy

It is evident that the intervention had a positive effect on pupils' performance. Pupils rated fake videos and manipulated images as less credible after the intervention, and the intervention did not lead them to rate credible news as less credible, which has previously been noted as a risk that is linked to exposure to fake news [69]. However, only pupils using digital tools in the post-test rated credible news as more trustworthy than before the intervention. This highlights the importance of using digital tools to support the process of verifying digital news, in line with theories of technocognition and transliteracy. Evidently, more pupils used digital tools after the intervention, but many pupils still did not.

The intervention did not have an overall effect on pupils' rating of true items as more true; instead, the mean score indicates that most pupils landed on the fence between 'not at all credible' and 'extremely credible'. This calls for further research to identify how to help pupils become better at identifying credible news as more than just 'somewhere in between'.

### 4.2. Variations across Countries, Performances and Attitudes

In relation to our research questions: (Q1) how does performance on pre- and post-test vary across countries? and (Q2) how do media attitudes and digital information attitudes vary across countries?, there are some interesting results. We did not conduct any direct analysis of pupils' self-rated attitudes and their performance, but we did see some interesting variations across the four countries that may, in a non-direct way, help us to understand the impact of the intervention, and the lack thereof.

The fact that the impact on performance was stronger in Romania and Spain than in Sweden and France may be due to the different skills of pupils participating in the educational intervention. Swedish pupils scored better than other pupils on the pre-test (M = 12.7), while Romanian pupils scored the lowest on the pretest (M = 10.4). In the post-test, Romanian pupils scored marginally better than Swedish pupils ($M = 13.3$ versus $M = 13.1$, respectively). Pupils in Spain also started with lower scores and gained more from this intervention than pupils in France and Sweden. This result may be understood in the light of our previous research, highlighting the great challenge facing education in Romania due to the media culture [68]. It may also be understood in relation to the fact that pupils in Romania and Spain, in particular, stated that they followed their gut feeling significantly less after the intervention. Previous research highlights how reliance on gut feelings and emotions may increase the belief in misinformation [73,74]. The positive impact in Spain may also be a result of pupils in Spain learning to see the importance of investigating sources of information. In addition, we found that pupils in three out of four countries learned to pay more attention to the design of images. This could also partly explain the better post-test performance in Spain and Romania, although pupils in Sweden did not seem to benefit from this. Thus, perhaps the combination of not relying on your gut feelings and paying attention to the design is key. The lack of impact for Swedish pupils may be understood in light of the fact that Swedish pupils self-rated as very skilled at fact-checking and searching information, which previous research has highlighted as problematic attitudes in relation to performance [13]. Further, we found that pupils in Romania were more positive overall to the digital tool than other pupils, particularly those in France and Sweden. Perhaps this also increased their engagement in using digital tools for fact-checking than other pupils. The more positive attitudes towards the digital tool in Romania and Spain may be an important element in the mix of attitudes and actions that made them navigate better in the post-test.

### 4.3. Limitations

This study has some important limitations. The small number of teachers and pupils in each country makes it difficult to generalise our results. The results might have been quite different in other groups in the same countries. Larger-scale studies across different types of schools would help us to better understand the extent to which the differences depend on the local setting. The study also holds important qualitative limitations, since we did not closely follow how pupils made sense of the tool and the educational design. A more detailed study of pupils' use of InVID-WeVerify could provide a better understanding as to why some pupils learn to navigate in more clever ways and change their attitudes, while other pupils do not. A potential limitation to this study is that the plug-in has not been designed with pupils in mind, but rather with professional fact-checkers. The statistically non-significant results in improving credibility assessment of French and Swedish pupils may be explained by the fact that these pupils gave the lowest scores to the InVID-WeVerify plug-in being elegant, exciting, and motivating (see Table A12 in the Appendix C). Usability, as well as the perceived utility, of an application has repeatedly been shown to be obfuscated by aesthetic preferences (e.g., [75]), and it has been shown to be a strong predictor of users overall impression [76]. However, aesthetics are valued differently cross-culturally [77], which may explain the differences in preferences of the plug-in between France and Sweden on the one hand, and between Romania and Spain on the other, and which is also mirrored in the impact of the tool on credibility assessment performance between the two groups of

countries. One potential avenue to investigate is to adapt the plug-in for pupils to improve its use in curricular activity.

## 5. Conclusions

In conclusion, we have conducted a novel study using a professional verification tool in contextually varied educational settings. The study delivered promising results, highlighting how it is possible to use a state-of-the-art image and video verification tools in classrooms and how this may support pupils to debunk fake images and videos. The intervention stimulated pupils' media and information literacy and made an impact on their ability to determine the credibility of visual fake news. The intervention also seemed to inspire pupils to rely less on gut-feelings. We find the fact that pupils in Romania benefited from this intervention particularly promising, since it indicates that the intervention may be especially useful for pupils facing special challenges with regard to misinformation. The participants in the present study were more likely to make use of digital tools post intervention and they generally performed better after the intervention. Thus, our research highlights that it is possible to implement advanced digital tools and stimulate pupils' knowledge, skills, and attitudes in an era of disinformation. Therefore, this study paves the way for future educational research on how to support pupils' lateral reading, technocognition, and transliteracy in different educational contexts, which is evidently possible with a combination of new technologies and teaching.

**Author Contributions:** Conceptualization, D.F.-M. and T.N.; methodology, M.G. and T.N.; formal analysis, M.G.; investigation, D.F.-M. and T.N.; resources, D.F.-M. and T.N.; writing—original draft preparation, T.N.; writing—review and editing, C.-A.W.A., D.F.-M. and M.G.; visualization, C.-A.W.A. and M.G.; supervision, D.F.-M.; project administration, D.F.-M.; funding acquisition, D.F.-M. and T.N. All authors have read and agreed to the published version of the manuscript.

**Funding:** This study is part of the YouCheck! project, funded by the European Commission programme Media Education for All, 2019–2020 (http://project-youcheck.com/ (accessed on 30 April 2021) and http://www.savoirdevenir.net/youcheck (accessed on 30 April 2021)). The study was also partly funded by Vinnova, grant number 2018-01279.

**Institutional Review Board Statement:** The study was conducted according to the guidelines of the Declaration of Helsinki.

**Informed Consent Statement:** Informed consent was obtained from all subjects involved in the study.

**Data Availability Statement:** We have not made data available.

**Acknowledgments:** We would like to thank the YouCheck! group for the involvement in designing and executing the study. We would also like to thank the participating teachers and pupils in the respective countries.

**Conflicts of Interest:** The authors declare no conflict of interest.

**Appendix A. Examples of Test-Items**

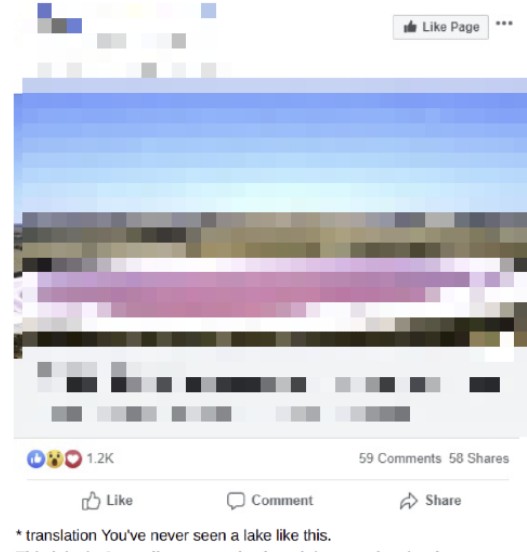

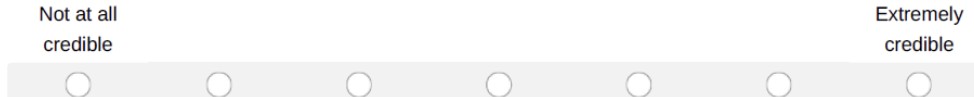

**Figure A1.** Viral, true example.

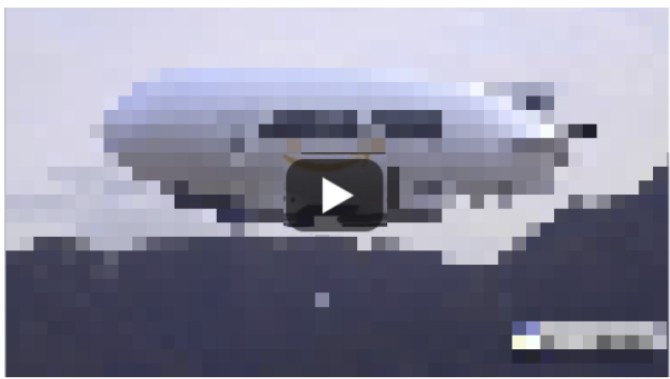

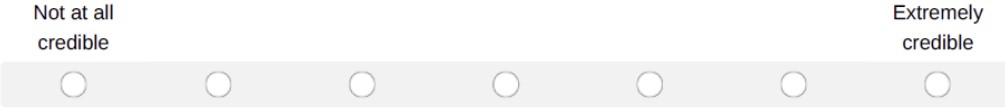

**Figure A2.** Fake video example.

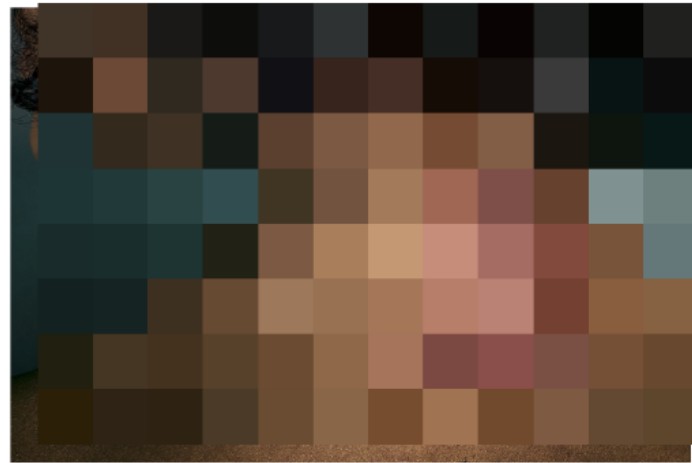

*5. "Intimate partner violence is one of the most common forms of violence against women and includes physical, sexual, and emotional abuse and controlling behaviours by an intimate partner" states WHO. Do you perceive the image above as credible evidence of intimate partner violence?

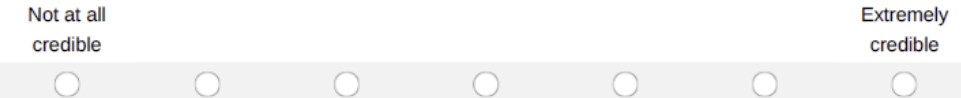

**Figure A3.** Manipulated image example.

**Appendix B. Background and Evaluative Questions**

*Appendix B.1. Questions about Background, Attitudes towards Digital News, Perceptions of Credibility and Digital Fact-Checking Tools*

1. What is your gender? (Man/Woman/Non-binary/Other alternative/Unsure/Prefer not to answer)
2. Do you follow news in multiple languages? (Yes/No)
3. How skilled are you at finding information online? (Very good—Very poor)
4. How skilled are you at critically evaluating online information? (Very good—Very poor)
5. How much of the information on the internet do you perceive as credible? (None—All)
6. How important is it for you to consume credible news? (Not at all important—Very important)
7. How do you discern factually correct information from information that is false? Please evaluate each of the following statements, on a scale from 1 (never) to 7 (very often)

   - I rely on journalist's reputation
   - I rely on news source/brand's reputation
   - I search for the source of the information
   - I compare different news sources to corroborate the facts
   - I consult factchecking websites in case of doubt
   - I check that the design of images and/or videos have good quality
   - I check what people say about the story online (e.g., on blogs, social media, opinion makers' websites)
   - I rely on my own knowledge and/or expertise on the subject
   - I rely on my gut feeling
   - I use digital tools (reverse image search, tineye, etc.)
   - I confront my impressions with friends and peers

*Appendix B.2. Additional Questions after the Intervention*

8.  Please rate the use of InVID-WeVerify, on various dimensions, on a scale from 1 (Not at all) to 7 (Very much):

    - enjoyable
    - pleasing
    - attractive
    - friendly
    - fast
    - efficient
    - practical
    - organized
    - understandable
    - easy to learn
    - clear
    - exciting
    - valuable
    - motivating
    - creative
    - leading edge
    - innovative

9.  Will you use digital tools like InVID-WeVerify in the future to fact-check online information? (Definitely not—Definitely)

## Appendix C. News Evaluation Attitudes

Means and standard deviations for pre- and post-test on news evaluation attitude test on pre- and post-test with separate estimates for each language are presented in Tables A1–A11. The rating scale ranged from 1 (never) to 7 (very often).

**Table A1.** I rely on journalists' reputation.

| Language | Pre | | Post | |
|---|---|---|---|---|
| | *M* | *SD* | *M* | *SD* |
| French | 3.1 | 1.9 | 3.5 | 1.8 |
| Romanian | 2.9 | 2.0 | 3.1 | 2.1 |
| Spanish | 2.6 | 1.8 | 3.4 | 2.1 |
| Swedish | 3.5 | 1.4 | 3.0 | 1.4 |

**Table A2.** I rely on my gut feeling.

| Language | Pre | | Post | |
|---|---|---|---|---|
| | *M* | *SD* | *M* | *SD* |
| French | 3.6 | 1.8 | 3.4 | 2.0 |
| Romanian | 4.0 | 2.0 | 1.9 | 1.7 |
| Spanish | 4.3 | 1.9 | 3.1 | 2.3 |
| Swedish | 3.4 | 1.7 | 3.0 | 1.6 |

**Table A3.** I use digital tools (reverse image search, tineye, etc.).

| Language | Pre | | Post | |
|---|---|---|---|---|
| | *M* | *SD* | *M* | *SD* |
| French | 5.0 | 1.8 | 4.5 | 1.7 |
| Romanian | 4.4 | 1.8 | 4.1 | 1.8 |
| Spanish | 4.9 | 1.6 | 4.8 | 1.6 |
| Swedish | 3.8 | 2.0 | 4.1 | 1.9 |

**Table A4.** I rely on news source/brand's reputation.

| Language | Pre | | Post | |
|---|---|---|---|---|
| | *M* | *SD* | *M* | *SD* |
| French | 5.0 | 1.6 | 4.4 | 2.0 |
| Romanian | 4.8 | 1.9 | 4.9 | 1.8 |
| Spanish | 5.0 | 1.9 | 5.3 | 1.6 |
| Swedish | 4.3 | 1.7 | 4.4 | 1.5 |

**Table A5.** I search for the source of the information.

| Language | Pre | | Post | |
|---|---|---|---|---|
| | *M* | *SD* | *M* | *SD* |
| French | 4.6 | 1.9 | 4.5 | 2.0 |
| Romanian | 5.2 | 1.7 | 5.7 | 1.3 |
| Spanish | 3.9 | 2.1 | 4.9 | 1.8 |
| Swedish | 4.5 | 1.7 | 4.4 | 1.8 |

**Table A6.** I compare different news sources to corroborate the facts.

| Language | Pre | | Post | |
|---|---|---|---|---|
| | *M* | *SD* | *M* | *SD* |
| French | 4.5 | 1.7 | 4.4 | 1.9 |
| Romanian | 5.3 | 1.8 | 5.7 | 1.6 |
| Spanish | 4.5 | 2.0 | 5.0 | 1.8 |
| Swedish | 5.3 | 1.7 | 5.4 | 1.5 |

**Table A7.** I check that the design of images and/or videos have good quality.

| Language | Pre | | Post | |
|---|---|---|---|---|
| | *M* | *SD* | *M* | *SD* |
| French | 2.6 | 1.8 | 3.1 | 1.9 |
| Romanian | 4.2 | 2.1 | 5.5 | 1.7 |
| Spanish | 2.6 | 1.8 | 3.7 | 2.1 |
| Swedish | 3.7 | 2.1 | 4.9 | 1.7 |

**Table A8.** I consult fact-checking websites in case of doubt.

| Language | Pre | | Post | |
|---|---|---|---|---|
| | *M* | *SD* | *M* | *SD* |
| French | 4.2 | 1.8 | 4.6 | 1.8 |
| Romanian | 5.3 | 1.5 | 5.5 | 1.7 |
| Spanish | 4.3 | 1.8 | 5.0 | 1.7 |
| Swedish | 4.2 | 1.8 | 4.1 | 2.0 |

**Table A9.** I confront my impressions with friends and peers.

| Language | Pre | | Post | |
|---|---|---|---|---|
| | *M* | *SD* | *M* | *SD* |
| French | 4.4 | 1.6 | 4.4 | 1.8 |
| Romanian | 4.6 | 1.9 | 4.9 | 1.9 |
| Spanish | 3.8 | 1.8 | 3.8 | 1.7 |
| Swedish | 4.6 | 1.7 | 4.5 | 1.6 |

**Table A10.** I check what people say about the story online (e.g., on blogs, social media, opinion makers' websites).

| Language | Pre | | Post | |
|---|---|---|---|---|
| | *M* | *SD* | *M* | *SD* |
| French | 4.6 | 1.7 | 4.5 | 1.5 |
| Romanian | 4.5 | 1.7 | 4.0 | 2.2 |
| Spanish | 4.9 | 1.5 | 4.8 | 1.7 |
| Swedish | 4.2 | 1.9 | 4.4 | 1.8 |

**Table A11.** I rely on my own knowledge and/or expertise on the subject.

| Language | Pre | | Post | |
|---|---|---|---|---|
| | *M* | *SD* | *M* | *SD* |
| French | 2.8 | 1.7 | 2.7 | 2.0 |
| Romanian | 2.9 | 1.8 | 3.2 | 2.2 |
| Spanish | 2.8 | 1.9 | 2.4 | 2.0 |
| Swedish | 2.6 | 1.5 | 2.5 | 1.4 |

**Table A12.** Means and Standard Deviations for InVID-WeVerify attitudes on post-test with separate estimates for each language. The rating scale ranged from 1 (not at all) to 7 (very much).

| Attitude | French | | Romanian | | Spanish | | Swedish | |
|---|---|---|---|---|---|---|---|---|
| | *M* | *SD* | *M* | *SD* | *M* | *SD* | *M* | *SD* |
| Appealing | 4.8 | 1.4 | 5.8 | 1.2 | 4.8 | 1.5 | 4.3 | 1.6 |
| Clear | 4.6 | 1.5 | 5.6 | 1.4 | 4.4 | 1.6 | 4.5 | 1.7 |
| Creative | 4.4 | 1.6 | 5.4 | 1.6 | 5.4 | 1.2 | 4.3 | 1.6 |
| Cutting edge | 4.5 | 1.5 | 5.4 | 1.4 | 5.8 | 1.2 | 4.4 | 1.5 |
| Easily learned | 4.5 | 1.5 | 5.7 | 1.3 | 5.1 | 1.7 | 4.5 | 1.7 |
| Efficient | 5.4 | 1.4 | 5.7 | 1.4 | 5.6 | 1.3 | 5.0 | 1.6 |
| Elegant | 4.0 | 1.5 | 5.7 | 1.4 | 4.7 | 1.5 | 4.0 | 1.5 |
| Exciting | 4.3 | 1.6 | 4.5 | 2.0 | 4.4 | 1.8 | 4.0 | 1.7 |
| Fast | 4.9 | 1.4 | 5.8 | 1.4 | 5.1 | 1.5 | 4.8 | 1.7 |
| Friendly | 4.5 | 1.6 | 5.7 | 1.2 | 4.4 | 1.9 | 4.5 | 1.5 |
| Innovative | 5.3 | 1.5 | 6.0 | 1.3 | 6.0 | 1.0 | 4.3 | 1.7 |
| Motivating | 3.9 | 1.6 | 4.6 | 1.8 | 4.3 | 1.8 | 3.8 | 1.6 |
| Practical | 5.1 | 1.4 | 5.8 | 1.5 | 5.7 | 1.5 | 5.1 | 1.6 |
| Simple | 4.4 | 1.3 | 5.9 | 1.3 | 4.9 | 1.6 | 4.5 | 1.7 |
| Unambiguous | 4.6 | 1.5 | 5.7 | 1.4 | 4.6 | 1.6 | 4.3 | 1.7 |
| Unorganised | 4.9 | 1.5 | 6.1 | 1.0 | 4.9 | 1.7 | 4.5 | 1.6 |
| Valuable | 5.1 | 1.6 | 5.4 | 1.6 | 5.7 | 1.5 | 4.9 | 1.6 |

**Table A13.** Detailed report of the total score entered into a 2 (using digital tools, between-subjects) ×4 (language) ×2 (time, within-subjects) mixed factorial ANOVA. Significant effects are in italics.

| Effect | SS | df | MSE | F | p | Partial $\eta^2$ |
|---|---|---|---|---|---|---|
| *Use of digital tools (1)* | 87 | 1 | 11.94 | 7.29 | 0.007 | 0.031 |
| Language (2) | 44 | 3 | 11.94 | 1.22 | 0.31 | 0.016 |
| Error (subject) | | 229 | | | | |
| *Time (3)* | 116 | 1 | 6.14 | 18.89 | 0.000 | 0.076 |
| 1 × 3 | 14 | 1 | 6.14 | 2.32 | 0.13 | 0.010 |
| *2 × 3* | 54 | 3 | 6.14 | 2.95 | 0.033 | 0.037 |
| Error | | 229 | | | | |

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
