# Peer review of "Combatting Visual Fake News with a Professional Fact-Checking Tool in Education in France, Romania, Spain and Sweden"

_information, doi:10.3390/info12050201_

Round 1
Reviewer 1 Report
- The topic is relevant and sufficiently described and well written
- Perhaps it would be appropriate to establish the concept of deep fake as an element of future risk
- It would have been interesting to share data regarding the misinformation of the countries chosen for the study, to learn about possible cultural biases.
- It would be interesting to know the reasons for the choice of these four countries and not others.
- Stimulating and interesting experiment to learn how little tech-mediated literacy pills can help better assessment and verification
- I think that the use of the concept of language instead of the country of origin is not sufficiently well described in figure 6. I do not quite understand the language / country correlation in the framework of the research, it should perhaps be clarified especially by including countries with existing of different official languages.
- The work is rigorous and provides an adequate and well-founded vision.
Author Response
- Perhaps it would be appropriate to establish the concept of deep fake as an element of future risk
--- We note that deepfake is a challenge in line with other misinformation (see Barari, S., Lucas, C., & Munger, K. (2021). Political Deepfake Videos Misinform the Public, But No More than Other Fake Media) and our results do not provide underpinnings of a separate analysis of deepfake.
- It would have been interesting to share data regarding the misinformation of the countries chosen for the study, to learn about possible cultural biases.
--- Yes, this would be interesting to know but difficult to assess. And we believe it would be slightly out of the scope of this paper.
- It would be interesting to know the reasons for the choice of these four countries and not others.
--- This was a joint EU project with countries included with different media cultures and we simply used the home countries of the participating researchers for convenience
- I think that the use of the concept of language instead of the country of origin is not sufficiently well described in figure 6. I do not quite understand the language / country correlation in the framework of the research, it should perhaps be clarified especially by including countries with existing of different official languages.
--- We use language as the measure throughout the paper, we focus on the language the tests were translated into. We have tried to make this more clear in the text.
Reviewer 2 Report
Fact-checking tools ARE available to the general public in some countries (l. 33).
Transliteracy also examines how ideas transfer from one format to another.
Small N so it is hard to generalize. With so many teachers, it's surprising that N is so small -- were whole classes participating or were there just small groups of students? -- I would call this an exploratory study.
How were the teachers and countries chosen?
Figure 1 is useful: design phrase seems logical. Seems like a limited assessment. What about impact over time? What was the time frame between lesson 1 and lesson 2?
Self-reporting is not a very accurate method.
It would be useful on p. 11 to display the statistics in tables.
Good that you noted limitations. I noted a couple of others.
How did you choose the examples? Students' prior knowledge (and culture) could also impact results.
Author Response
Fact-checking tools ARE available to the general public in some countries (l. 33).
--- We have toned down the rhetoric
Transliteracy also examines how ideas transfer from one format to another.
--- Transliteracy has many aspects and we base our theoretical considerations on the work of Frau-Meigs highlighting how there are multiple digital formats, in addition to print-based formats.
Small N so it is hard to generalize. With so many teachers, it's surprising that N is so small -- were whole classes participating or were there just small groups of students? -- I would call this an exploratory study.
--- The number of teachers in the focus groups are higher than in the intervention because the set up of the intervention was more complicated than conducting the interviews. Interviews were conducted before the Covid-19 pandemic. The interventions were set up only with teachers finding time and possibilities to implement the educational design in their teaching in person during the Covid-19 pandemic. The study is partly exploratory when it comes to investigating the differences between countries. We added "exploratory" when presenting our research questions to make this clearer. For the hypotheses of overall performance, we do not agree that the number of participants is small. The effects are clear across countries.
How were the teachers and countries chosen?
--- By convenience as part of a project including countries from the European North, East and South, this was a study formulated in an EU project and the countries were simply the countries of the participating researchers.
Figure 1 is useful: design phrase seems logical. Seems like a limited assessment. What about impact over time? What was the time frame between lesson 1 and lesson 2?
We did not conduct a follow up so we cannot say to what extent the impact lasted over time. The lessons were conducted on the same day or days following, depending on the schedule of each school.
Self-reporting is not a very accurate method.
--- As we wanted to understand the participants' own perception and attitudes towards online information, self-reporting is the only relevant way of measuring. And skills to navigate misinformation was measured by test-items.
It would be useful on p. 11 to display the statistics in tables.
--- We have now redistributed statistics into tables and figures and cleaned some of the text up as not to duplicate data and make the text somewhat more readable.
Good that you noted limitations. I noted a couple of others.
--- Would you care to elaborate on these noted limitations?
How did you choose the examples? Students' prior knowledge (and culture) could also impact results.
--- Test-items were chosen to cover different challenges of misinformation – with an emphasis on items with misleading images and videos. The examples did not relate to any of the countries regarding the content of the information. Using items in pre- and post-tests in a within subject design was a way for us to safeguard that students’ prior knowledge may impact the results. In the discussion we note how culture may play a role that we are unable to capture fully in our design study.
Reviewer 3 Report
Dear authors,
The submitted manuscript has very good intentions and although it is a topic of interest to the scientific community, I do observe some specific corrections to be made. I comment on them below:
- What are the future lines of research proposed by the work?
-What does it contribute to the subject area compared to other published materials?
-Are the conclusions consistent with the evidence and arguments presented? Do they address the main question posed?
- Expand the conclusions and connect it in relation to the results obtained so that they present more forcefulness.
- Incorporate and update bibliographic references in the text and at the end of it. Some are recommended:
doi.org/10.1080/10584609.2019.1668894
doi.org/10.3390/educsci11040144
-What have been the limitations of the methodology? You must express them in the manuscript.
Author Response
- What are the future lines of research proposed by the work?
--- We have already proposed two important aspects for future research:
"This calls for further research to identify how to help pupils become better at identifying credible news as more than just `somewhere in between'."
"One potential avenue to investigate is to adapt the plug-in for pupils to improve its use in curricular activity."
-What does it contribute to the subject area compared to other published materials?
--- We added the following line to make the contribution more clear: "we have conducted a novel study in that we used a professional verification tool in an educational setting"
-Are the conclusions consistent with the evidence and arguments presented? Do they address the main question posed?
--- Yes
- Expand the conclusions and connect it in relation to the results obtained so that they present more forcefulness.
--- We are not quite sure what is meant here with "more forcefulness", we have clearly stated hypotheses and research questions which we then address in our discussion in relation to the results.
- Incorporate and update bibliographic references in the text and at the end of it. Some are recommended:
doi.org/10.1080/10584609.2019.1668894
doi.org/10.3390/educsci11040144
--- We have provided the most up-to-date references relevant to secondary school education in relation to digital literacy. These provided references are not relevant to our study.
-What have been the limitations of the methodology? You must express them in the manuscript.
--- We have already included a section of limitations. If you mean only specifically the methodology, a classroom intervention is a well-established methodology in educational research with its pros and cons, but such a methodological discussion seems out of the scope of this particular paper. Please feel free to elaborate on any particular limitations you have considered.
Round 2
Reviewer 2 Report
line 237-238: repeated word doesn't clarify, try "All items were news intended to intended ...
What do the numbers on the test mean (is it a Likert scale? What does 1 mean .. up to 7?). Who determines the number of points -- and how (the basis)? Same question for table 4. Self-rating is not very reliable. You should mention Stanford's civic reasoning research. Your literature review is still limited.
Author Response
Many thanks for useful comments and feedback. We have now updated the paper in line with your recommendations to increase the quality of our paper.
Point 1: Limited research review. “You should mention Stanford's civic reasoning research. Your literature review is still limited.”
Response 1: In our paper we already discuss how their work is an important starting point for our study and refer to the following studies conducted by the Stanford group:
Breakstone, J.; Smith, M.; Wineburg, S.; Rapaport, A.; Carle, J.; Garland, M.; Saavedra, A. Students’ civic online reasoning: a national portrait. Report, 2019.
McGrew, S.; Ortega, T.; Breakstone, J.;Wineburg, S. The Challenge That’s Bigger than Fake News: Civic Reasoning in a Social Media Environment. American Educator 2017, 41, 4.
McGrew, S.; Smith, M.; Breakstone, J.; Ortega, T.; Wineburg, S. Improving university students’ web savvy: An intervention study 2019. 89, 485–500. doi:10.1111/bjep.12279.
McGrew, S. (2020). Learning to evaluate: An intervention in civic online reasoning. Computers & Education, 145, 103711. doi:https://doi.org/10.1016/j.compedu.2019.103711
McGrew, S., & Byrne, V. L. (2020). Who Is behind this? Preparing high school students to evaluate online content. Journal of Research on Technology in Education, 1-19. doi:10.1080/15391523.2020.1795956
Wineburg, S., & McGrew, S. (2019). Lateral Reading and the Nature of Expertise: Reading Less and Learning More When Evaluating Digital Information. Teachers college record, 121(11), 1-40.
We have now also added recently published studies.
Point 2: line 237-238: repeated word doesn't clarify, try "All items were news intended to intended ..
Response 2: We have now clarified this by replacing the repeated “item” with “social media post”
Point 3: What do the numbers on the test mean (is it a Likert scale? What does 1 mean .. up to 7?). Who determines the number of points -- and how (the basis)? Same question for table 4. Self-rating is not very reliable.
Response 3: The rating scales come from previous research and we use the same five-point rating scales as in previous research. Questions in Table 4 have been used to highlight how self-reported attitudes relate to skills in Nygren & Guath 2019, 2021. Questions regarding fact-checking have previously been discussed in Fruzanu & Corbu (2020) and Ştefăniţă, O., Corbu, N., & Buturoiu, R. (2018) with a seven-point rating scale which comes from an official EU initiative. See Appendix B1 with all the questions asked and response options. We have now emphasised this more in the text (lines 307-313).
We agree that self-reports hold important limitations, but they may reflect attitudes among participants and indicate for instance over-confidence among participants when correlated with test-results (Nygren & Guath, 2019). Mixing test-items and self-reported questions is necessary to identify relationships between perceived skills and actual performance (Nygren & Guath, 2019, 2021; Porat et al 2018).
Reviewer 3 Report
Dear authors,
The submitted manuscript has improved with the corrections made, but I still see some weaknesses. I comment on them below:
-Some questions to answer are: What is the main question addressed by the research? Is it relevant and interesting?
-What does it contribute to the subject area compared to other published materials?
-The methodology used does not have sufficient consistency to be published in a high impact journal
-Indicate how the results obtained are related or not to the literature and the opinion of prominent authors on the subject.
- It is recommended to expand the conclusions that summarize the results obtained.
-Connect the conclusions in relation to the results obtained so that they present greater forcefulness.
-Are the conclusions consistent with the evidence and arguments presented? Do they address the main question posed?
- The discussion and conclusions are not related to the main objective.
- It is recommended to update the bibliography.
Author Response
Many thanks for useful comments and feedback. We have now revised our paper in line with your recommendations to increase the quality of our paper.
Point 1: What is the main question addressed by the research? Is it relevant and interesting?
Response 1: As we already note in the paper, today this type of research is highlighted as very important by national and international organizations, and more importantly there is a call from researchers to investigate “more intensive digital literacy training models (such as the “lateral reading” approach used by professional fact checkers)’ (Guess et al. 2020). In addition we find it most relevant to add to research that has highlighted how novices should learn to use tools and cognitive strategies more in line with experts. Our findings are also interesting in light of a previous call from McGrew for more research on civic online reasoning with new and more complex tasks and test-items paying attention to pupils' knowledge, skills and attitudes in different educational settings. We have now addressed this by highlighting the call for more research with a few more references. How teenages across cultural and national borders may develop technocogntion and transliteracy is certainly a relevant and interesting topic in a world saturated by online misinformation.
Point 2: What does it contribute to the subject area compared to other published materials?
Response 2: When researchers call for educational interventions to support strategies like technocogntion, transliteracy and civic online reasoning - we test this in real world classrooms - adding new insights deriving from the complexity of practice across national borders.
Point 3: The methodology used does not have sufficient consistency to be published in a high impact journal.
Response 3: Design-based research is a well-established methodology which is often published in high end educational journals (See for instance Anderson & Shattuck 2012 Design-Based Research: A Decade of Progress, Educational Researcher). We have now added a more detailed description about design-based research that inspired us in our study.
Point 4: Indicate how the results obtained are related or not to the literature and the opinion of prominent authors on the subject. It is recommended to expand the conclusions that summarize the results obtained. Connect the conclusions in relation to the results obtained so that they present greater forcefulness. Are the conclusions consistent with the evidence and arguments presented? Do they address the main question posed? The discussion and conclusions are not related to the main objective. It is recommended to update the bibliography.
Response point 4: We have now updated the introduction, methods section, results and discussion. Please find edits and updates especially on lines 48-64 (emphasising our main objective), 87-100 (the relevance of our research), 151-178 (the methodology) and 407-537 (tying discussion to main objective and results) and in the bibliography. We have chosen to keep the conclusions brief and concise, however we have now extended the Discussion section to better highlight how our findings relate to previous research on the subject and a clearer connection to the main objective. The bibliography has now been extended with more references related to recent media and information literacy studies as well as design-based research.
Round 3
Reviewer 2 Report
thank you for your changes
Author Response
Thank you for your help in improving our manuscript
Reviewer 3 Report
Dear authors,
The submitted manuscript has improved with the latest corrections, although I still see some issues for improvement. I comment on them below:
- It is recommended to synthesize section 1 Introduction, it is excessively long and reiterates concepts
- Expand the conclusions and connect it in relation to the results obtained so that they present more forcefulness.
-Are the conclusions consistent with the evidence and arguments presented? Do they address the main question posed?
- Incorporate and update bibliographic references in the text and at the end of it. Some are recommended:
doi.org/10.3390/su12052123
doi.org/10.3390/educsci11040144
Author Response
Thank you for your valuable comments and input. Please find our replies to your concerns below.
Point 1: It is recommended to synthesize section 1 Introduction, it is excessively long and reiterates concepts
Reply to point 1: We added the lines 178-181 to synthesise the introduction and make it clear that our focus lies on 1) external threat of misinformation, 2) cognitive challenges for pupils an 3) the existence of digital fact-checking tools to combat these obstacles with the help of design-based research for an educational intervention.
Point 2: Expand the conclusions and connect it in relation to the results obtained so that they present more forcefulness.
Reply to Point 2: The conclusions have been somewhat extended to better accommodate our findings (lines 616-617 and 620-626), however, we are careful not to over-analyse and speculate on our findings.
Point 3: Are the conclusions consistent with the evidence and arguments presented? Do they address the main question posed?
Reply to Point 3: We believe they are and that they do
Point 4: Incorporate and update bibliographic references in the text and at the end of it. Some are recommended:
doi.org/10.3390/su12052123
doi.org/10.3390/educsci11040144
Reply to Point 4: We extended our bibliography quite extensively in the previous update. We do not find the provided references relevant for our study and will therefore not include them.